# Using Fuzzy Cognitive Maps to Assess Liveability in Slum Upgrading Schemes: Case of Pune, India

**Subhashree Nath** [1] and **Raphael Karutz** [2,*]

1   DLGS—Dresden Leibniz Graduate School, IOER—Leibniz Institute of Ecological Urban and Regional Development, 01217 Dresden, Germany; s.nath@dlgs.ioer.de
2   UFZ—Helmholtz Centre for Environmental Research, Department of Urban and Environmental Sociology, 04318 Leipzig, Germany
*   Correspondence: raphael.karutz@ufz.de; Tel.: +49-341-235-4771

**Abstract:** Liveability assessments of informal urban settlements are scarce. In India, a number of slum upgrading schemes have been implemented over the last decades aiming at better living conditions. However, these schemes rarely consider improvement in liveability as an explicit criterion, assuming that better physical conditions and the provision of basic services inevitably lead to better liveability. We use Fuzzy Cognitive Maps (FCMs) to analyse liveability in four different informal settlements in Pune (India). We compare the liveability by conducting semi-structured interviews with residents and by analysing them in individual and aggregated FCMs. Each settlement represents an archetypical form of the upgradation process: *non-upgraded* (base case), *in-situ upgraded*, *relocated*, and *temporary resettlement*. The FCMs show that the liveability indicators *availability of community space*, *proximity to public transportation*, *feeling of belonging*, and *good relationship with neighbours and community* are central elements of these neighbourhoods' liveability. The results suggest that upgradation may lead to an improved overall liveability but can also reduce it if not designed properly. The fostering of community agency, an integration of the neighbourhood into the formal city fabric, and the maintaining of cohesion during the shift from horizontal to vertical living emerged as critical factors. To ensure sustainable integration of liveability considerations in slum upgrading schemes, we suggest using indicators well-adapted to the local context, co-created with local experts and stakeholders, as well as periodic post-occupancy liveability evaluations.

**Keywords:** liveability; informal settlements; slums; slum upgrading; fuzzy cognitive maps; post-occupancy evaluations

## 1. Introduction

The United Nations Statistical Department states that as of 2018, 23.5% of the total global urban population was living in slums, defined as densely populated urban areas, generally informally inhabited by people of low-income groups, and characterized by substandard living conditions [1,2]. In India, the label slum is often used synonymously with the umbrella term 'informal housing', including 'unauthorised housing', 'notified slums', 'recognised slums', 'identified slums' and 'unidentified slums'. These settlement typologies however, vary significantly in terms of property rights and legalities. The research paper only considers notified slums, referring to a settlement notified as a slum in the Indian State Government's official gazette under the applicable Slum Act, qualifies them for upgrading [2].

Of these 23.5%, Central and Southern Asia account for 22.7% [3]. In India alone, 377 million people live in slums according to the latest Census [4]. The most prevalent response to existing slums has been slum upgrading, which is considered socially and financially most appropriate, compared to other historic approaches, like slum eradication through forced eviction and displacement, relocation to a new housing outside city limits, or providing public housing [5] (pp. 12–15). Slum upgrading is the process of gradually

improving, formalizing, and incorporating informal settlements into the city [5] (p. 16). While there are several approaches to improve the physical infrastructure of a slum, from in-situ rehabilitation to relocation, there is limited knowledge about the post-upgrading liveability, especially in comparison to the various approaches. Most global liveability rankings and surveys like the *Economist Intelligence Unit (EIU)*, the *Mercer Quality of Living Survey*, and the *Organization for Economic Cooperation and Development (OECD) Better Life Index (BLI)* are based on formal settlements in developed countries, considering categories like socio-political stability in terms of civil unrest, crimes, socio-cultural environment like the discomfort of climate to travellers, personal freedom, social support system, etc. [6–9]. Although these categories are fitting for developed countries, there is a knowledge gap in understanding how liveability perceptions vary in Low/Middle-Income Countries (LMIC) like India, where a large percentage of the population lives in informal settlements and often struggle even to access basic services [10,11].

For informal settlements, it is often assumed that simply ensuring safer housing structures and providing basic services like household-level water, sanitation, and electricity, inevitably leads to an improved quality of life. Thus, slum upgrading schemes rarely consider improvement in liveability as a separate criterion, but a by-product of upgrading physical infrastructure. Post-occupancy studies of these upgrading schemes report, however, that residents are often dissatisfied with their living conditions despite the upgrades [12–14], and there is a prevalent 'rebound phenomenon' where occupants abandon the upgraded houses and move back to the slums [15]. This further establishes the importance of a liveability analysis of these schemes, both before and after they are implemented. However, most liveability ranking and indices evaluate liveability performance by quantifying urban qualities, for example, SDG indicator 11.2.1 which describes the "proportion of the population that has convenient access to public transport, by sex, age and persons with disabilities" [16]. Similarly, all the 79 liveability indicators prescribed by the Ministry of Urban Development, Govt. of India, are expressed primarily in statistical metrics. For example, one of the indicators under the category of education is mentioned as "Percentage of the school-aged population enrolled in schools" [17]. While the percentage is relevant for assessing development, such purely quantitative expressions fail to capture the causal effects of different factors which contribute to the overall liveability experience. We argue that a neighbourhood-based liveability analysis centred on the resident's experience is more appropriate. We consequently define liveability as the quality of life in a certain community, measured by the resident's satisfaction with the living environment, safety, attractiveness, crime rate, education and employment opportunities, social cohesion, inclusion, and amount of open space [18,19].

The study addresses the research question of how liveability can be assessed in Pune's Slum Upgrading Schemes from a resident-centred perspective.

To answer this, a mixed-method approach is developed which involves using Fuzzy Cognitive Maps (FCMs) based on semi-structured interviews with residents from slums and upgraded neighbourhoods. FCMs map the cause-and-effect relations between defined variables [20]. Each variable has an everyday meaning and hence, the developed map model is easy to understand, even by a non-technical audience, allowing discussion among non-technical participants, like residents of the settlements. The integration of FCM and liveability analysis offers an academic contribution to the municipality decision support discipline, along with facilitating community engagement in upgrading schemes for better liveability. As a proof-of-concept, the method is applied to find the key indicators that residents consider most influential for a better liveability both in slums and in the neighbourhoods upgraded under the different Schemes of Pune City.

This paper is structured as follows: After a short overview of the case study context, history of India's slum upgrading policies and a description of the methodology developed for this study, suitable liveability indicators for informal settlements are derived and FCMs are conceptualized. Based on interviews from archetypical (post-) upgradation settlements in Pune, FCMs are drawn and analysed and discussed as a proof-of-concept, showing their

potential application in supporting resident-centric decision-making to prioritize actions for improving liveability in a neighbourhood.

### 1.1. Background

Pune is the second-largest city in the state of Maharashtra, 150 km from the state capital, Mumbai, and the eighth largest Urban Agglomeration (UA) of India with a population of approx. 5 million people [21,22]. The paper focuses on Pune City (Figure 1), which is part of the UA and has a population of approx. 3.1 million [23]. Estimates on the number of slum dwellers vary: while Pune Municipal Corporation (PMC) [24] assumes 1.2 million slum dwellers (38.7%) for the year 2009, the national census estimates the slum population at 0.69 million for the same year [4]. Despite these differences in slum population estimates, it is irrefutable that a large share of Pune residents live informally. The "Revised City Development Plan (CDP) of Pune City-2041" [25] depicts the vision of a "Slum Free City with Inclusive and affordable Housing for all" [25] (p. 89). This is to be reached by applying two major schemes: "Basic Services for Urban Poor (BSUP)" which is a submission of "Jawaharlal Nehru Urban Renewal Mission (JNURM)", and "Rajiv Awas Yojana (RAY)". RAY suggests state governments a two-step approach for improving the living conditions in existing slums and preventing the growth of further informal settlements. The first step is a curative strategy suggesting up-gradation, redevelopment, or resettlement, while the second is a preventive strategy, which includes developing sufficient accessible affordable housing through cross-subsidies and incentives [25] (p. 90). The curative strategy uses the "Tenability Analysis" shown in Figure 2. The land on which the slum pockets are located are assessed to determine whether they should be (a) rehabilitated through resettlement in a new area, (b) upgraded in-situ via retrofitting, or (c) redeveloped, i.e., upgraded through shifting to a multi-storey housing within the same land parcel. For this research paper, we consider these three approaches and the corresponding neighbourhood to study how the residents perceive liveability, and which liveability indicators they consider most relevant. As a reference, a fourth neighbourhood of slums which has received no major upgrading is also considered. The selected neighbourhoods are discussed further in the Section 3. The next sub-section discusses the process of contextualizing liveability indicators, to be adopted in the consecutive steps for liveability analysis.

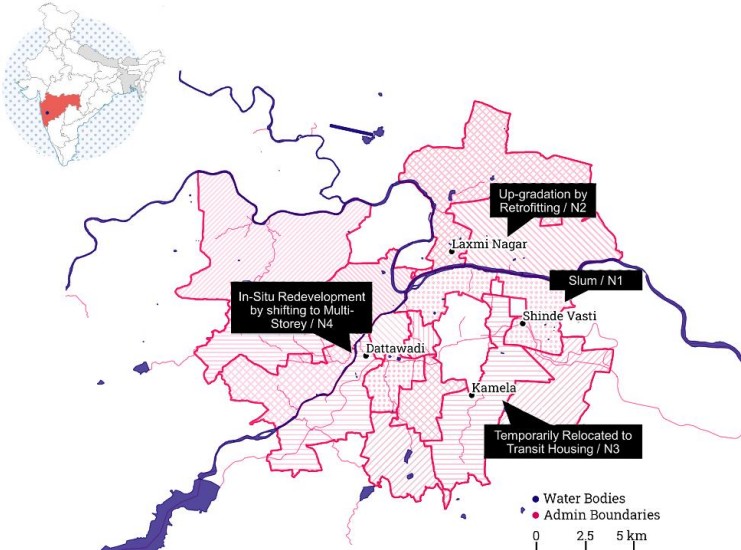

**Figure 1.** Pune City and the location of the four slum neighbourhoods investigated (authors' rendering based on Maps of India [26] and PMC [27].

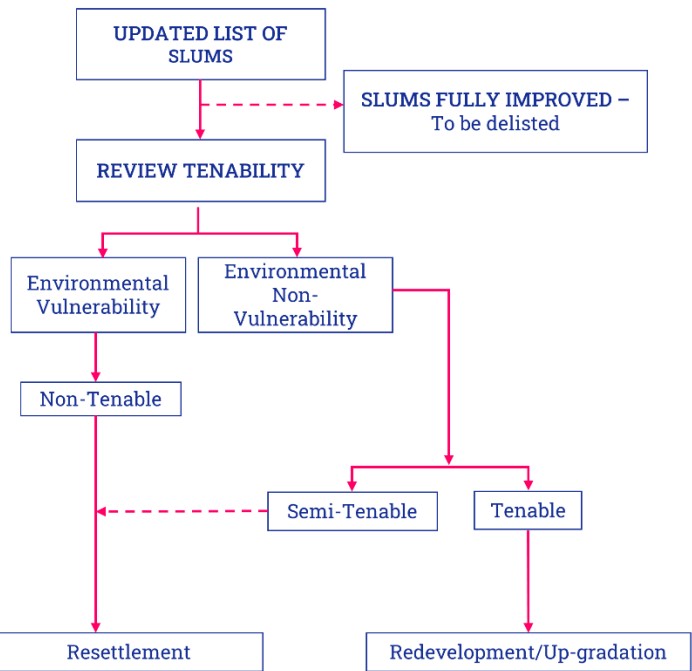

**Figure 2.** Tenability Analysis recommended in Rajiv Awas Yojana (PMC, 2012) (Redrawn by author).

### 1.2. Liveability Indicators

There is limited guidance about what constitutes a liveable city from a Low- and Middle-Income Country's (LMIC) perspective [28]. Goal 11 of the SDG, Sustainable Cities and Communities, comprises the targets of ensuring access for all to adequate, safe and affordable housing with basic services, upgrading slums, providing universal access to safe, inclusive, accessible, green, and public spaces [29]. While the New Urban Agenda (NUA) mentions enhancing liveability for all [30], liveability analysis of slums or informal settlements, either in their original condition or post-upgrading, are not yet common. Hence, there is a need to contextualize liveability indicators for slums and analyse how the upgrading policies affect these indicators.

The study adopts 13 indicators (Figure 3) for analysing slums and slum upgrading schemes, based on the current literature on liveability indicators which considers a neighbourhood scale, through the analysis of residential preference, from an LMIC perspective [28,31–33]. The indicators have been classified under four thematic dimensions: Physical, Social, Functional, and Safety. Given the complexity and subjectivity of the concept of liveability, this list of indicators does not attempt to be comprehensive, however, the thematic grouping aims to address the potentially relevant aspects in the context of informal settlements.

The *physical dimension* refers to the resident's immediate environment and how it contributes to fulfilling the fundamental human need for providing shelter. The indicators aim to cover the resident's perception of various aspects of this environment like access to basic infrastructure, public and green space, quality, and maintenance of the infrastructure. This dimension is particularly relevant in the case of slums since the absence of one or more of these indicators is characteristic of slums [34]. Upgradation of the physical infrastructure is one of the primary goals of the rehabilitation schemes under BSUP [35] and hence, it can be assumed that the inhabitants of the rehabilitated settlements will score higher on indicators from the physical dimension. Yet, a recent study by Debnath et al. [15] highlights that despite the physical improvement, rehabilitated occupants move back to slums, chiefly due to financial distress and built-environment related discomfort.

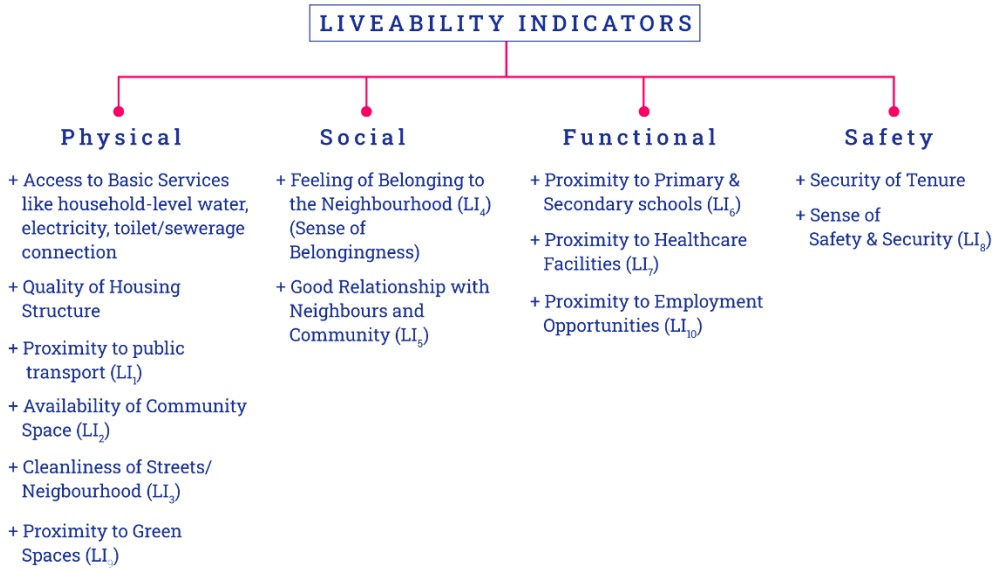

**Figure 3.** Liveability Indicators (LIs) contextualized to LMIC neighbourhoods.

The *social dimension* indicates the community ties and social connection of the settlement, which contribute to a sense of belonging to the neighbourhood. Nijman [36] states that slum rehabilitation is an institutional change, where an entire informal built environment is shifted to a formal housing structure. His further research shows that for at least 70% of residents of Dharavi slums, a sense of community is the most valued aspect [37]. Features of the built environment in the rehabilitated housing have a significant influence on the sense of community, often leading to social isolation if the design restricts access to communal spaces, as confirmed by the study conducted by Debnath et al. [15]. They also note that the situation is more pronounced for women, who used to use outdoor open spaces in the slums for social consumption, as they performed most of the household activities outdoors while socialising with neighbours. The study results show that 80% of the respondents are distressed due to lack of outdoor spaces, daylight availability in the corridors and in between the buildings, insufficient cooking spaces, etc., making it difficult for them to socialize even with neighbours [13]. This further affects their relationship with neighbours which is one of the indicators contributing to the social dimension.

The *functional dimension* covers the functionality provided by the resident's environment and is assessed by the proximity and access to services and opportunities. It represents how the physical environment (covered by the indicators under the physical dimension) can provide for the inhabitants and how they can use it for bettering their lives [33]. Holt-Jenson (2001) further mentions that the functional dimension consists of indicators implying that the sense of well-being depends on good provision and location of communication systems, shops, kindergartens, shopping centres, clinics, schools, and other services (as cited in [31]). It implicitly refers to the location of the settlements, which could affect the proximity and hence access to employment opportunities, education, and healthcare. Rehabilitation through relocation is often done in sites which are in the urban periphery, leading to increased travel time and cost, which often leads to occupants moving back to informal settlements closer to the city centre where most opportunities are located [5].

The *safety dimension* addresses the neighbourhood's sense of security in terms of both legal status and safety from crime or hazards. It is one of the basic needs, reflected in the fact that everybody wants to live in a crime-free, safe neighbourhood [31]. The absence of property rights can be translated into tenure insecurity and is one of the characteristics of slums [2]. Rehabilitated occupants have tenure security and might initially perceive a higher sense of safety from both natural and manmade hazards due to improved physical infrastructure, which follows the standard by-laws and safety norms, along with the new

'legal' status. However, a weaker sense of community and relationship with neighbours may lead to a reduced sense of security as neighbours can be considered an important resource in preventing crime [38]. Farahani [38] further mentions that neighbourhoods that have a high degree of social interaction can help in controlling crime informally through surveillance of a neighbour's home and looking out for strangers, often called 'eyes on the street', a phrase popularized by Jacobs [39].

It must be noted that the classification of indicators under various dimensions is not definitive nor mutually exclusive. For example, indicators measuring proximity could fit in both physical and functional dimensions, since the location of these facilities is a physical aspect but their accessibility and proximity effects their functionality. Additionally, not all indicators contribute equally to liveability and some indicators may have a higher influence than others. Further, there are often external factors influencing the performance of certain indicators like *Feeling of Belonging to the Neighbourhood*, which might vary with the changing generations of inhabitants. A neighbourhood which fosters a sense of belonging for an older couple might not do the same for a younger one.

These correlations between different indicators and the varying perceptions of their functionality makes a compelling case for the use of FCM to analyse liveability.

## 2. Materials and Methods

We interpret liveability as the quality of life in a certain community, measured by the resident's satisfaction with a defined set of indicators. Consequently, a method is developed centred around the residents' perception of liveability. It attempts to find how liveability perception changes as slums are upgraded and how the liveability indicators (LI, Figure 3) influence each other, by interviewing the residents. While the initial plan was to conduct in-person interviews, due to the onset of the COVID-19 pandemic, this was changed to telephonic interviews. The interviews were analysed using Fuzzy Cognitive Maps to detect the indicator(s) which holds the highest influence, by mapping the causal relationship between the LI. The method developed was tested on four selected neighbourhoods of Pune (cf. Figure 1):

a.　Neighbourhood 1—*Shinde Vasti*: Slum with no intervention.
b.　Neighbourhood 2—*Laxmi Nagar*: Slum upgraded by retrofitting.
c.　Neighbourhood 3—*Kamela*: Slum temporarily relocated to transit housing. This is considered a substitute for upgrading by relocation since no interviewees could be contacted in a relocated neighbourhood and the transit housing share similar characteristic of a relocated housing.
d.　Neighbourhood 4—*Dattawadi*: In-situ redevelopment by shifting to multi-storey housing.

The particularities of each case study site are described in detail in the Results Section. The rest of the section describes the steps of the method in their order of execution.

### 2.1. Semi-Structured Interviews

To find the causal relationship between the selected indicators, semi-structured telephone interviews were conducted with residents of the respective neighbourhoods. The interviews consisted of a structured part in which questions on liveability were asked following a pre-developed questionnaire, and an open part, in which the residents were asked further questions on their neighbourhood and their everyday life in the slums. These open questions were not systematically included in the development of the FCMs but helped for a better overall contextual understanding of the situation in the respective neighbourhoods and were taken up in the discussion of results.

The questionnaire was developed iteratively, with several rounds of pilot interviews, during which four residents from three different neighbourhoods were interviewed. Although the pilot neighbourhoods did not correspond to the four neighbourhoods finally selected, the pilot interviews were instrumental in resolving critical issues such as choice of words and phrasing, questionnaire format, length, and rating system. The question-

naire consisted of three parts (see supplementary material for questionnaire, individual responses, and corresponding FCMs).

a. The first question elicits the interviewees' general perception about their neighbourhood, whether they are satisfied, or whether they think it can be improved, or are dissatisfied. To minimize language barriers during the interviews, the wording chosen for this three-point scale answers were "*a lot*" (satisfied), "*a bit*" (can be improved), and "*not at all*" (dissatisfied).

b. The second set of questions finds how they rated the performance of the 13 selected liveability indicators, with the same three-point scale. Indicators which are inherently improved during the upgrading process, like the Q*uality of Housing, Access to Basic Amenities,* and *Security of Tenure* were left from the next round of questions and mapping since their performance has objectively improved through the upgrading. Of the remaining indicators, the ones which were not rated satisfactory were taken forward to check how their performance can be positively influenced by other indicators. This limits the number of causal relationships which is crucial to limit the time taken for each interview. When time is not a constraint, finding a causal relationship between all indicators would be preferred for optimal results.

c. The third set of questions attempts to identify the interlinkages between the indicators, the potential of indicators to improve those that were rated either "*not at all*" (unsatisfactory) or "*a bit*" (can be improved), based on the interviewees' perception. Depending on whether the indicators were thought to have some influence in improving the unsatisfactory indicators, they could be rated from "*no influence*" (influence value 0) to "*very little*" (influence value = 0.3), "*a bit*" (influence value = 0.6), and "*a lot*" (influence value = 1.0).

A total of 11 telephonic interviews were conducted: three each from Neighbourhood 1, 2, and 3, and two from Neighbourhood 4. On average, the interviews lasted for 25 min. In addition to analysing the interviews using FCM, basic spatial analysis using Google Earth Satellite Images was done to identify the distribution of schools, employment opportunities, healthcare facilities and parks within a radius of 400 m and 800 m. The radius was based on the "walkable catchments" concept [40] where an area of 400 m radius is within 5 min walking distance and 800 m within 10 min. For more in-depth information on Pune's individual slums and their morphology, as well as socio-economic status, see the extensive mapping by NGOs Maharashtra Social Housing and Action League (MASHAL) [41] and Shelter Associates [42].

### 2.2. Fuzzy Cognitive Maps (FCMs)

The study understands liveability as inherently a social phenomenon, an outcome of how humans experience their environment and the degree to which this environment facilitates their daily activities, their social mobility, and interaction with one another. As such, it is suitable to use fuzzy logic to analyse liveability which uses human-like reasoning for a better representation of reality. Fuzzy Cognitive Maps combine fuzzy logic and cognitive mapping to develop a network of cause and effect relations between different variables (liveability indicators) the values of which are "verbally described and do not have to be dimensionally defined" [43]. The cause-and-effect relations show the positive or negative influence of the concepts on each other with verbally assigned weights [43] such as "*very little*", "*a bit*", or "*a lot*". These verbally described causal relations are mapped to a fuzzified value between 0 and 1, transforming qualitative values to quantitative values. The model is appropriate for representing unstructured knowledge since it is not limited by exact values and measurements of the variables [44]. The possibility of aggregating individual maps prepared by different interviewees adds to the utility of FCMs [45,46] in facilitating community engagement and participatory decision-making processes.

### 2.2.1. Components of an FCM

An FCM has the following components (cf. Figures 4 and 5) [20,43,45,47–49]:

1. *Concepts (from here on as Liveability Indicators (LI))*: They represent the drivers/indicators that have influence (causation) on the system in consideration and can be represented with $LI_1$, $LI_2$ ... $LI_n$. They can be defined contextually and need not have a dimensional definition.
2. *Directed edges*: Arrows with signs (+/−) depicting the relationships between concepts (causality), indicating that a concept causes another concept. A positive correlation '+' between $LI_1$ and $LI_2$ means increasing $LI_1$ increases $LI_2$ and decreasing $LI_1$ decreases $LI_2$ while the reverse is the case for a negative correlation '−'.
3. *Weight of directed edge*: While the directed edges or arrows with signs show a causal relation between two indicators, the weight (between 0 and 1) shows the degree to which one indicator causes another. The stronger the causation, negative or positive, the closer the value is to 1 and the weaker the causation, the closer the value is to 0.
4. *Adjacency Matrix*: Mathematical representation of the FCM to analyse the centrality of a concept or indicator (conceptual centrality) and the role of each component in the network, whether it is ordinary, driver/transmitter or receiver.

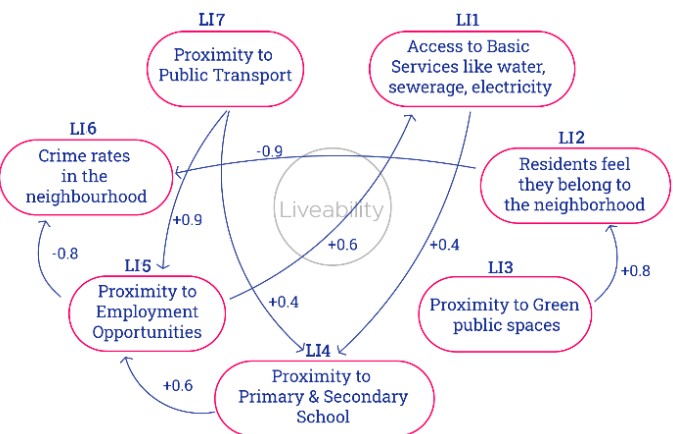

**Figure 4.** An example of an FCM describing the causal relation between a few Liveability Indicators (denoted by $LI_1$, $LI_2$ ... $LI_7$). The presence of an arrow between two LI indicates a causal relation where the LI at the base of the arrow causes LI at the head of the arrow. The '+' or '−' sign along with their values indicate the degree of causation. For example, an increase in access to education opportunities ($LI_4$) can cause an increase (+0.6) in access to employment opportunities ($LI_5$). Access to employment ($LI_5$) has a negative causation (−0.8) on the crime rate, which decreases with the increase in employment opportunities. The values, +0.6 and −0.8 show the strength or possibility of causation. Closer the value to 1, regardless of sign, stronger its causation power. Therefore, $LI_4 \rightarrow LI_5$ has relatively weaker causation than $LI_5 \rightarrow LI_6$.

An *ordinary indicator* is both affected by, and influences, other indicators.
A *driver/transmitter indicator* only influences other indicators.
A *receiver indicator* is only affected by other indicators.
*Centrality* determines the importance of an indicator in the overall network.

There are several centrality measures which can be used in network analysis, e.g., Degree Centrality, Betweenness Centrality, Closeness Centrality. In this study, we use Degree Centrality (DC), which is the absolute sum of an indicator's in-arrows (Indegree, ID) and out-arrows (Outdegree, OD), see Equation (1).

$$DC_{LIi} = \sum_{j=1}^{N} |ID_{ij}| + \sum_{k=1}^{N} |OD_{ik}| \tag{1}$$

Degree Centrality is a good pointer of the contribution of an indicator in an FCM, which shows how connected the indicator is to other indicators, and the cumulative strength of these connections [20], making it a priority action point. Further, looking at the

indegrees and outdegrees of a central indicator, it can be found whether the indicator is mainly getting influenced (Receiver) or is influencing another indicator (Driver) or both (Ordinary).

|     |                                                              | LI1  | LI 2 | LI3 | LI4  | LI 5 | LI6  | LI7 |
|-----|--------------------------------------------------------------|------|------|-----|------|------|------|-----|
|     | LI1: Access to basic services like water, sewerage, electricity | 0    | 0    | 0   | +0.4 | 0    | 0    | 0   |
|     | LI2: Residents feel they belong to the neighborhood           | 0    | 0    | 0   | 0    | 0    | -0.9 | 0   |
| A=  | LI3: Proximity to green public spaces                         | 0    | +0.8 | 0   | 0    | 0    | 0    | 0   |
|     | LI4: Proximity to Primary & Secondary School                  | 0    | 0    | 0   | 0    | +0.6 | 0    | 0   |
|     | LI5: Proximity to Employment Opportunities                    | +0.6 | 0    | 0   | 0    | 0    | -0.8 | 0   |
|     | LI6: Crime rates in the neighbourhood                         | 0    | 0    | 0   | 0    | 0    | 0    | 0   |
|     | LI7: Proximity to Public Transport                            | 0    | 0    | 0   | +0.4 | +0.9 | 0    | 0   |

**Figure 5.** Adjacency Matrix for FCM in Figure 5. For example, $LI_1$–$LI_4$ = +0.4 is the causal edge value, the causality indicator $LI_1$ imparts on $LI_4$.

### 2.2.2. Drawing and Aggregating FCMs

1.  Indicators not rated satisfactory (*"not at all" or "a bit"*) by the interviewees are taken forward to check how their performance can be improved by other indicators.
2.  To determine the interlinkages, the responses gathered from the questionnaire were transformed into respective FCMs using the software "Mental Modeler" and "FCM Expert".
3.  Degree Centrality is calculated for all indicators from each interviewee using the Adjacency Matrix.
4.  The FCMs of interviewees from each neighbourhood are aggregated into one, resulting in four neighbourhood level FCMs, which highlight all the indicators considered relevant by the interviewees of each neighbourhood. For aggregation, the total weight of each directed edge is calculated via matrix addition to derive a new aggregated adjacency matrix. All individual FCMs are weighted equally in the aggregation. The aggregated FCMs can be normalized by dividing the matrix elements by the number of individual interviews [20,50]. This has been done in Figure 15 for better comparability across the four slums.

### 3. Results

The following describes the key findings from the interviews and the analysis:

### 3.1. Neighbourhood 1: Shinde Vasti, Hadapsar: Informal Settlement with No Intervention

The first group of three interviewees resided in Shinde Vasti (Figure 6), an informal settlement in the Hadapsar ward which has not yet received any infrastructure upgrades from the municipality, although efforts towards providing basic amenities are in progress. All three interviewees have been living in Shinde Vasti for more than 20 years and each has incrementally constructed their houses which now have a *pucca* (permanent construction typically made of brick, concrete) construction.

In total, two out of three interviewees from Neighbourhood 1 rated their neighbourhood as *satisfactory* and one interviewee rated it as *unsatisfactory*. *Cleanliness of the Neighbourhood, Proximity to Public Transport, Proximity to Green Spaces*, and *Availability of Community Space* were recurrently rated unsatisfactory.

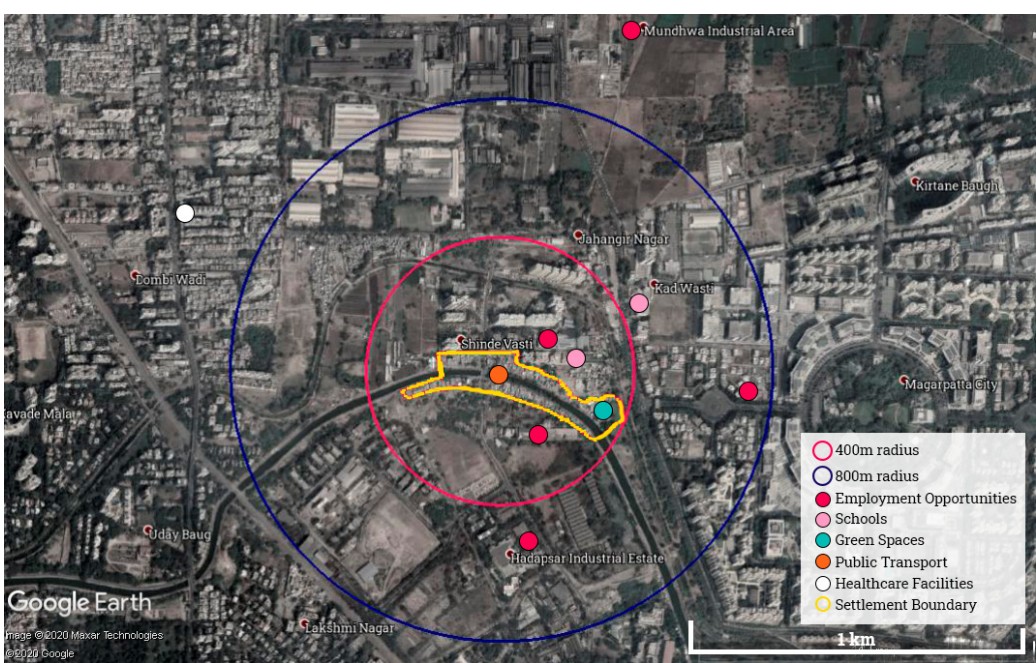

**Figure 6.** Shinde Vasti Satellite Image (author processing on the base map from Google Earth [51]).

Figure 7 shows an FCM drawn from one of the interviews from Neighbourhood 1. According to Interviewee 1, *Cleanliness of Neighbourhood (LI₃)*, would have a strong positive influence on *Availability of Community Space (LI₂)*, *Proximity to Primary and Secondary Schools (LI₆)*, and *Proximity to Green Spaces (LI₉)*. While an increase in *Feeling of Belonging (LI₄)* can positively influence LI₃ and lead to a cleaner neighbourhood. In this FCM, *LI₃: Cleanliness of Neighbourhood* has the highest Degree Centrality, which means LI₃ has a high contribution to the overall liveability and the concepts LI₂, LI₆, and LI₉ which are directly connected to LI₃ can be deduced as priority action points based on the respondent's perception.

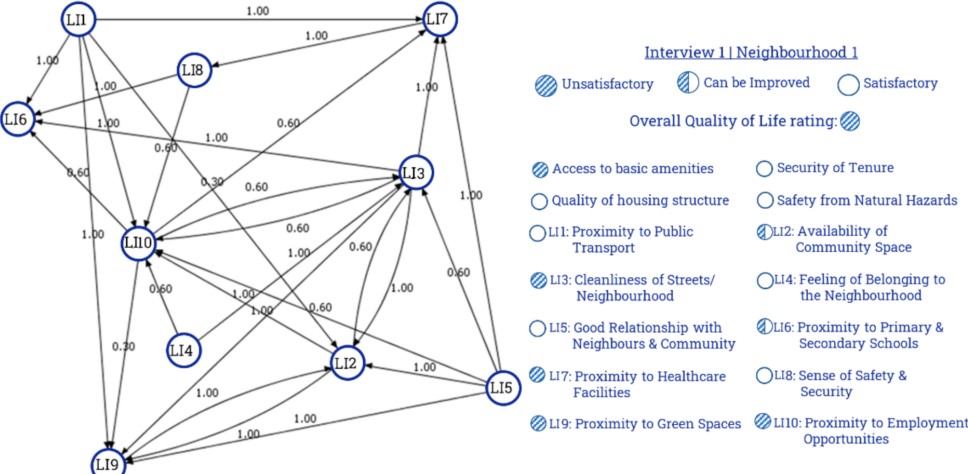

**Figure 7.** Individual FCM for Interviewee 1.

The aggregated FCM for Neighbourhood 1 (Figure 8) shows that *LI₂: Availability of Community Space* is the most central indicator followed closely by *LI₃: Cleanliness of Streets/ Neighbourhood*, which the narratives also confirm. The FCM also indicate a strong influence of *LI₅: Good Relationship with Neighbours and Community on LI₂: Availability of Community Space. Additionally, LI₁: Proximity to Transport* has been identified as a driver indicator, which only influences the other indicator and does not get influenced. This was evident from all

three interviews, where it was mentioned how improving LI$_1$ can lead to an improvement in *Proximity to Schools (LI$_6$)*, *Healthcare Facilities (LI$_7$)*, and *Access to Community Space (LI$_2$)*.

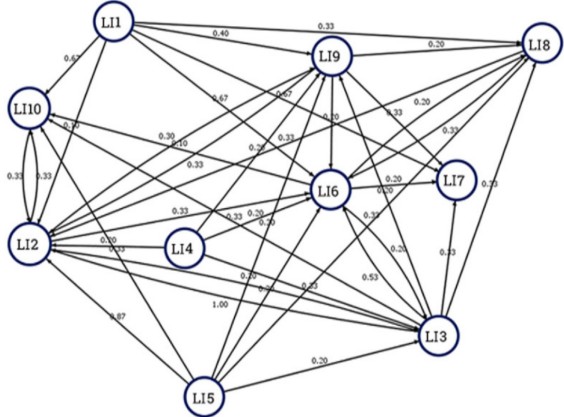

Neighbourhood 1: Informal Settlement with No Intervention

| Component | Indegree | Outdegree | Centrality | Type |
|---|---|---|---|---|
| LI2: Availability of Community Space | 3.16 | 1.16 | 4.33 | ordinary |
| LI3: Cleanliness of Streets/Neighbourhood | 0.93 | 2.73 | 3.66 | ordinary |
| LI6: Proximity to Primary and Secondary Schools | 2.46 | 0.7 | 3.16 | ordinary |
| LI1: Proximity to Public Transport | 0 | 2.83 | 2.83 | driver |
| LI9: Proximity to Green Spaces | 1.29 | 1.06 | 2.36 | ordinary |
| LI5: Good Relationship with Neighbours and Community | 0 | 2.13 | 2.13 | driver |
| LI 10: Proximity to Employment Opportunities | 1.76 | 0.33 | 2.09 | ordinary |
| LI8: Sense of Safety and Security | 1.39 | 0.66 | 2.06 | ordinary |
| LI7: Proximity to Healthcare Facilities | 1.53 | 0 | 1.53 | receiver |
| LI4: Feeling of Belonging to the Neighbourhood | 0 | 0.93 | 0.93 | driver |

**Figure 8.** Aggregate FCM and adjacency matrix for Neighbourhood 1.

### 3.2. Neighbourhood 2: Laxmi Nagar, Yerwada. Upgrading by Retrofitting

Laxmi Nagar (Figure 9) is an informal settlement in Yerwada, where the settlements are being gradually formalised by giving tenure rights and in-situ upgrading by retrofitting. The NGO MASHAL has been coordinating and managing these projects in Laxmi Nagar, Yerwada [52].

Two out of three interviewees rated their neighbourhood generally as *satisfactory*, and one interviewee answered that the neighbourhood *can be improved*. Despite the improvements in the service infrastructure and built quality of the houses, service and maintenance remains an issue. The residents also expressed their dissatisfaction with not having a park/public garden in the vicinity, especially since they could see the park doubling up as a community space.

The aggregated FCM (Figure 10) highlights that *Proximity to Public Transport (LI$_1$)* and *Green Spaces (LI$_9$)* are the most central indicators. It is also evident from the FCM that the residents expect an improvement of *LI$_1$* to benefit *LI$_9$*.

### 3.3. Neighbourhood 3: Kamela, Kondhwa. Transit Housing for SRA In-Situ Multi-Storey Housing

The Kamela Slum Rehabilitation Project (Figure 11) was initiated in 2017 when around 270 slums were demolished [53] and the occupants were shifted to a transit housing with a rent contract of 4 years. The interviews mentioned that they were supposed to shift to the redeveloped housing by August 2020, but the shift has been delayed because of the pandemic. Two out of three interviewees felt that the neighbourhood *can be improved*. They were *dissatisfied* regarding *Proximity to Employment Opportunities (LI$_{10}$)* due to dislocation, followed by non-*Availability of Community Space (LI$_2$)*. The aggregated FCM (Figure 12) shows that *Feeling of Belonging to the neighbourhood (LI$_4$)* has the highest centrality.

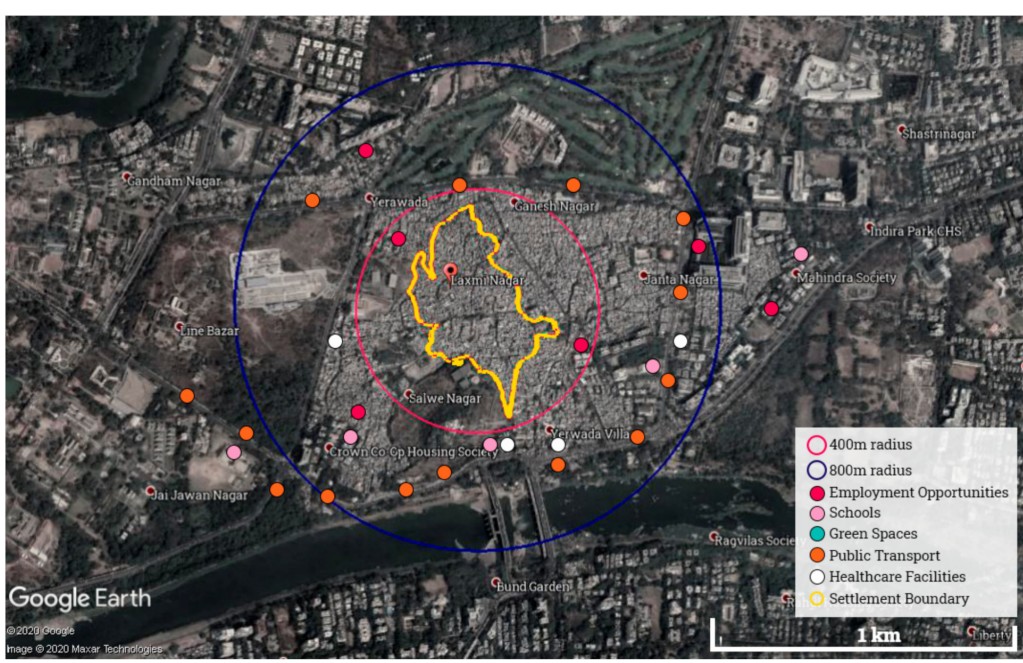

**Figure 9.** Laxmi Nagar, Yerwada (author processing on the base map from Google Earth [51].

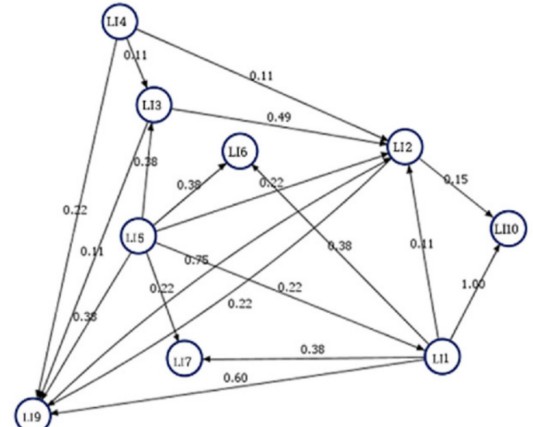

Neighbourhood 2: Upgrading by in-situ retrofitting

| Component | Indegree | Outdegree | Centrality | Type |
|---|---|---|---|---|
| LI1: Proximity to Public Transport | 0.22 | 2.46 | 2.68 | ordinary |
| LI9: Proximity to Green Spaces | 1.53 | 0.75 | 2.28 | ordinary |
| LI2: Availability of Community Space | 1.68 | 0.37 | 2.06 | ordinary |
| LI5: Good Relationship with Neighbours and Community | 0 | 1.79 | 1.79 | driver |
| LI 10: Proximity to Employment Opportunities | 1.15 | 0 | 1.15 | receiver |
| LI3: Cleanliness of Streets/Neighbourhood | 0.48 | 0.6 | 1.08 | ordinary |
| LI6: Proximity to Primary and Secondary Schools | 0.75 | 0 | 0.75 | receiver |
| LI7: Proximity to Healthcare Facilities | 0.6 | 0 | 0.6 | receiver |
| LI4: Feeling of Belonging to the Neighbourhood | 0 | 0.44 | 0.44 | driver |

**Figure 10.** Aggregated FCM and adjacency matrix for Neighbourhood 2: Proximity to Public Transport ($LI_1$) and Proximity to Green Space ($LI_9$) have the highest centrality, while Good Relationship with Neighbours and Community ($LI_5$) is a Driver Concept, having a strong influence on the network.

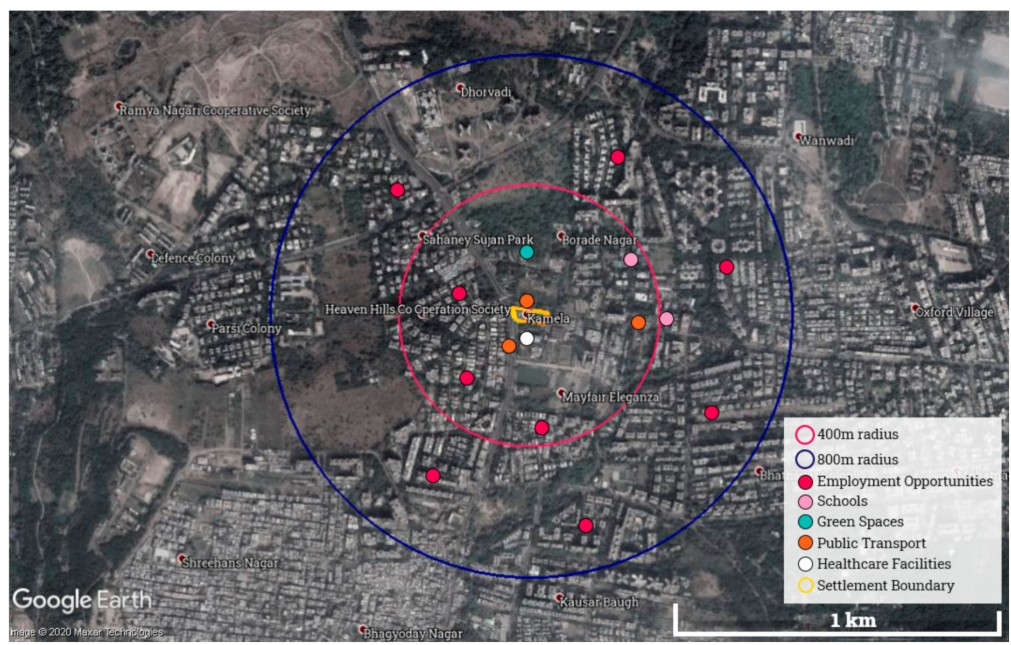

**Figure 11.** Kamela Rehabilitation Site (author processing on base map from Google Earth [51].

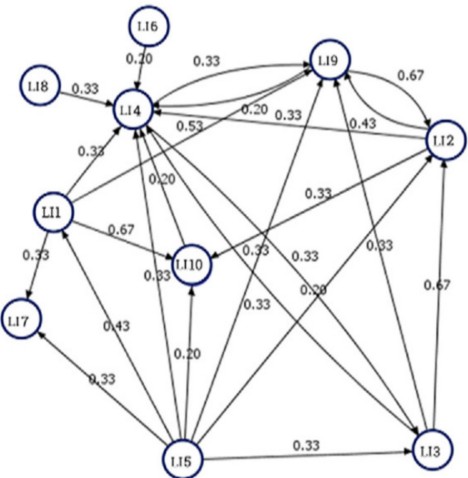

Neighbourhood 3: Transit Housing (relocated temporarily) for SRA in-situ Multi-Storey Housing

| Component | Indegree | Outdegree | Centrality | Type |
|---|---|---|---|---|
| LI4: Feeling of Belonging to the Neighbourhood | 2.26 | 0.66 | 2.93 | ordinary |
| LI9: Proximity to Green Spaces | 1.96 | 0.86 | 2.83 | ordinary |
| LI2: Availability of Community Space | 1.53 | 1.09 | 2.63 | ordinary |
| LI1: Proximity to Public Transport | 0.43 | 1.86 | 2.29 | ordinary |
| LI5: Good Relationship with Neighbours and Community | 0 | 2.16 | 2.16 | driver |
| LI3: Cleanliness of Streets/Neighbourhood | 0.66 | 1.33 | 1.99 | ordinary |
| LI 10: Proximity to Employment Opportunities | 1.2 | 0.2 | 1.4 | ordinary |
| LI7: Proximity to Healthcare Facilities | 0.66 | 0 | 0.66 | receiver |
| LI8: Sense of Safety and Security | 0 | 0.33 | 0.33 | driver |
| LI6: Proximity to Primary and Secondary Schools | 0 | 0.2 | 0.2 | driver |

**Figure 12.** Aggregated FCM and adjacency matrix for Neighbourhood 3.

### 3.4. Neighbourhood 4: Dattawadi SRA In-Situ Multi-Storey Housing

The redevelopment process of the Dattawadi slum (Figure 13) was initiated in 2012 and the beneficiaries shifted to the redeveloped multi-story (11 stories) housing in September 2016. The two residents who agreed to the interview have been residing in the slum for over 20 years and both were generally *satisfied* with the outcome of the redevelopment. While they were *satisfied* with the performance of physical infrastructure as well as the provisions made for social infrastructure like community halls, they suggested a drop in *Feeling of Belonging (LI$_4$)* and *Relationship with Neighbours/Community (LI$_5$)*. The aggregated FCM (Figure 14) also shows these indicators having the highest centrality and *Availability of Community Space (LI$_2$)* is perceived as a driver indicator.

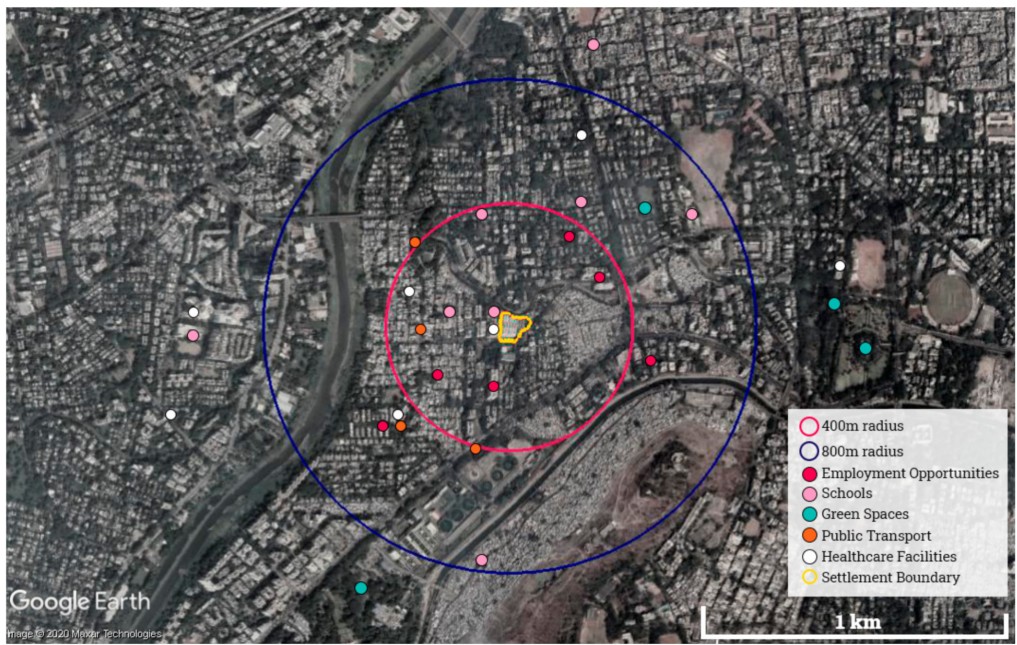

**Figure 13.** Dattawadi SRA Project Site (author processing on the base map from Google Earth [51].

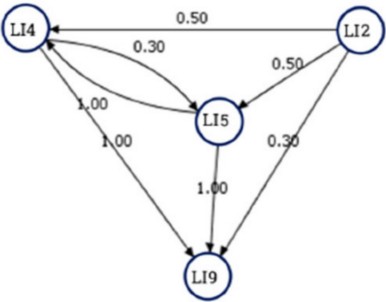

Neighbourhood 4: In-Situ Redevelopment to Multi-Storey Housing

| Component | Indegree | Outdegree | Centrality | Type |
|---|---|---|---|---|
| LI5: Good Relationship with Neighbours & Community | 0.8 | 2 | 2.8 | ordinary |
| LI4: Feeling of Belonging to the Neighbourhood | 1.5 | 1.3 | 2.8 | ordinary |
| LI9: Proximity to Green Spaces | 2.3 | 0 | 2.3 | receiver |
| LI2: Availability of Community Space | 0 | 1.3 | 1.3 | driver |

**Figure 14.** Aggregated FCM neighbourhood and adjacency matrix for Neighbourhood 4.

## 4. Discussion

The aggregated FCMs from the four neighbourhoods identify Availability of Community Space (LI$_2$, Physical dimension), Proximity to Public Transportation (LI$_1$, Physical dimension), Good Relationship with Neighbours and Community (LI$_5$, Social dimension), and Feeling of Belonging (LI$_4$, Social dimension) as the key (central) indicators (see Figure 15). Interestingly, these indicators were not always the ones rated to be least satisfactory. Due to their high centrality, however, they may function as important nodes in the system, driving other indicators and being influenced by them. Further elaborations by the respondents during the interviews confirmed the closely interconnected nature of the indicators and shed some light on the sometimes-complex causalities as perceived by the residents. A few aspects deserve particular attention:

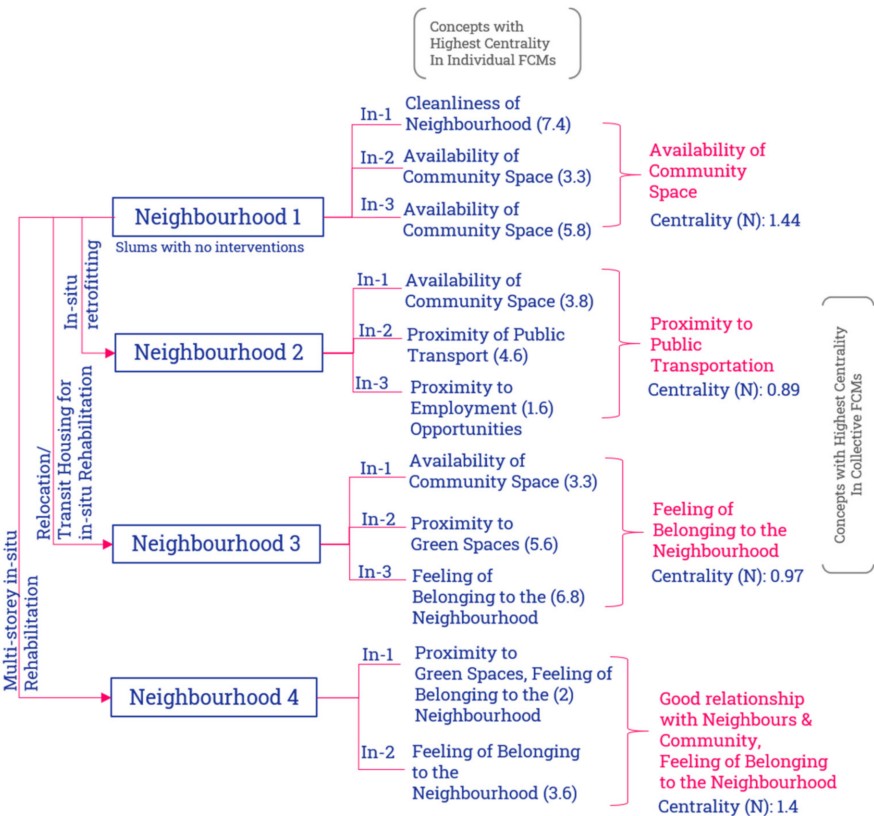

**Figure 15.** Summary of most central indicators from individual and aggregated FCMs. Individual interviews are numbered (In-1 corresponding to interview 1 of the respective neighbourhood, etc.). The centrality is provided in parentheses. For the aggregated centrality, values were normalized (divided by the number of interviews, indicated by the (N)) to allow for better comparability. For Neighbourhood 4, the two indicators *Good Relationship with Neighbours and Community* and *Feeling of Belonging to the Neighbourhood* have equal centrality and are thus both displayed.

In Neighbourhood 1, *Availability of Community Space* was explained to not only provide space for celebrations, fostering community spirit, but also to offer economic opportunities *(LI$_{10}$)*. One interviewee explained: "… *if we would have a community space, like a hall, it would give the ladies the space to come together and start some small business*". In absence of an actual community hall, public green spaces *(LI$_9$)* were seen as potential alternative meeting places *(LI$_2$)*, but only when clean of litter *(LI$_3$)*. Here, the interviewees saw a need for action to engage the community in positively shaping their neighbourhood themselves, instead of depending on external interventions. To achieve this, a *Good Relationship with Neighbours and Community (LI$_5$)* is found to be required. Such a collective agency is also found in literature to be critical in urban poverty reduction, improvement in infrastructure and delivery of

services from the municipalities [54]. The municipality and NGOs involved in upgrading should focus on capacity building within the community, to help form Community Based Organizations (CBO). A partnership between the government and the CBO could lead to the fast-tracked yet holistic development of such informal settlements.

In Neighbourhood 2, the high centrality of *Proximity to Public Transport (LI$_1$)* may be surprising at first glance, as many bus stops are in close vicinity. In fact, two of the three interviewed residents were satisfied with this indicator. However, the access to transportation was seen as critical for the access to public green spaces *(LI$_9$)*, which they were missing in their community, as well as employment opportunities *(LI$_{10}$)*. Not only does the spatial "nearness" to the next bus stops play a role here, as interviewees from other neighbourhoods confirmed, but also it is about the ease of traversing the distance, shaped by factors such as the bus schedule frequency and routes or the dirtiness of the path to the bus stop. The high prioritization of good access options by the residents points to a major concern in the case of in-situ upgradation of without enough measures to integrate the upgraded settlement into the network of infrastructures and services enjoyed by the other planned housing developments. Upgrading must represent a shift in the attitude of the local government, recognizing the rights of the inhabitants of the upgraded settlements.

Neighbourhoods 3 and 4, both characterized by a change from horizontal to vertical living, featured social indicators most centrally. Despite the generally positive perception of the two indicators *Feeling of Belonging to the Neighbourhood (LI$_4$)* and *Good Relationship with Neighbours and Community (LI$_5$)*, the interviewees from Neighbourhood 4 mentioned a deterioration resulting from the rehabilitation: *"before we used to live in a muhalla (community) and everybody lived close by . . . people used to keep their doors open. Now we have moved into flats, nobody keeps their doors open anymore . . . so nobody cares what happened outside of their door"*. In terms of perceived improvement in overall neighbourhood quality, three interviewees from Neighbourhood 3 and 4, described an overall improvement in the new housing due to the rehabilitation: *"Our new housing has everything . . . we also have a park"*, *"the bus stop is close by and it is also close to where I work"*. The other two interviewees perceive that it can still be improved but added that this is temporary. This generally positive impression *(LI$_1$, LI$_9$, LI$_{10}$)* is somewhat divergent to various findings of previous studies [12–15,55], which argue that residents are dissatisfied with their new housing, especially after residing there for a few years, and many say they prefer their previous living conditions. One possible explanation is that the current schemes of upgrading have learnt from the past drawbacks, and the more recent projects strive to be more participatory, considerate of the social habits of the residents, and thoughtful of the transitional impediments when residents have to shift from horizontal living to vertical living. These values are mandated in the Revised City Development Plan (CDP) of Pune City-2041 [25]. The recommendations mentioned in CDP, guiding various slum upgrading projects in Pune to achieve the vision and intention for a 'Slum-free India', are coherent with the recommendations on slum upgradation by UN-Habitat [5]. However, despite an emphasis on a 'bottom-up' approach and participatory design with the help of Community-Based Organisations (CBOs) and NGOs, the process lacks liveability assessments.

The interviews also revealed that despite being from the same neighbourhood, perceptions of liveability can be varied, if not polarising. In some cases, like that of Neighbourhood 1 and 2, the variation can be attributed to the spatially dispersed nature of the neighbourhood. While in general, the perception could also vary due to individual expectations, which has not been covered in this study. Subsequently, slum upgrading schemes must define their goal considering rehabilitation/upgradation as a series of incremental strategies, rather than a one-off infrastructure development project.

Based on the study conducted, the following recommendation can be adopted for improved liveability outcomes of slum upgrading schemes:

1. Contextualizing indicators: As pointed out in the introduction, liveability indicators for informal urban contexts are still scarce. Building on the indicators presented in this work, on-the-ground contextualization could greatly benefit the local applicability of

   liability indicators. This could be achieved through conducting workshops with field experts like NGOs, CBOs and local municipality, stakeholders, and academics.

2. Tripartite partnership: As our results show, community agency plays a vital role in successful rehabilitation. Enabling partnership between CBOs, NGOs and the municipality is important to vocalize resident concerns and ensure that built-environment upgradations consider the social habits of the neighbourhood.

3. Integration to the formal city fabric: Strategies need to be developed for comprehensive integration of the rehabilitated neighbourhoods to the formal city fabric, safeguarding access to the various functional attributes of liveability, like proximity and access to public transport, education, healthcare.

4. Mandating periodic liveability assessments: Credible before- and after rehabilitation evaluations are required to better capture the actual effect of the intervention. In particular, Post-Occupancy Liveability Evaluation (POLE) could ensure the workability of completed projects as well as gathering feedback on residents' change in liveability.

## 5. Conclusions

This study forwards a novel method to assess liveability perceptions in slums and upgraded neighbourhoods by identifying key indicators with the highest influence on liveability which can act as positive performance leverage. A resident-centric liveability analysis method was developed based on interviews and FCMs.

It was found that the indicators considered most influential by the residents for a better liveability were Availability of Community Space, Proximity to Public Transportation, Good Relationship with Neighbours and Community, and Feeling of Belonging. Integrating the method in the SWOT (Strength-Weakness-Opportunity-Threat) analysis, commonly done at the beginning of the design phase of rehabilitation projects, would help in prioritising actionable points while considering the outcome of improved liveability from the residents' perspective.

The study was limited by a small sample size and the telephonic nature of the interviews which inhibits the interviewee from responding with assurance, often in the apprehension of the intention of the interviewer. In telephonic interviews, the time the respondent is willing to dedicate is shorter, and there are more distractions than in face-to-face interviews. To make the best out of the interviews despite these limitations, the study was limited to the analysis of causal relationships of non-satisfactory indicators with the other indicators, rather than attempting to elicit causal relationships between all indicators. Another attempt to shorten the interview time was excluding indicators expected to be improved by the nature of the rehabilitation scheme (*Quality of Housing, Access to Basic Amenities,* and *Security of Tenure*). While effectively reducing interview times, these measures limit the scope of analysis, in particular with regards to potential paradox effects due to the upgradation process. For a follow-up investigation, we thus recommend a larger sample size, face-to-face interviews, and the inclusion of all potentially relevant indicators.

However, the study expanded the application of Fuzzy Cognitive Maps in resident-centric liveability assessments and decision guiding in slum upgrading schemes. It also opens the discourse towards the causal relation of liveability indicators that has not been explored in depth in the current literature. The foremost contribution of the study is the development of a replicable method that can collect qualitative inputs in terms of resident's perception of liveability in slums, quantify, aggregate, and analyse them using FCMs. It thus fills the knowledge gap in liveability studies in LMIC, as most liveability assessment methods are borrowed from the Global North, lacking the informality characteristic of most LMICs.

**Supplementary Materials:** The following are available online at https://www.mdpi.com/article/10.3390/urbansci5020044/s1.

**Author Contributions:** S.N. conceptualized the research and developed the methodology, curated, and analysed the data. S.N. prepared the original draft. R.K. supervised, reviewed, and edited the paper. All authors have read and agreed to the published version of the manuscript.

**Funding:** This work was conducted as part of the Belmont Forum Sustainable Urbanisation Global Initiative (SUGI)/Food-Water-Energy Nexus theme for which coordination was supported by the US National Science Foundation under grant ICER/EAR-1829999 to Stanford University. UFZ received funding from the Federal Ministry of Education and Research (BMBF) under grant 033WU002. Any opinions, findings, and conclusions, or recommendations expressed in this material do not necessarily reflect the views of the funding organizations.

**Institutional Review Board Statement:** Ethical review and approval were waived for this study, since no personal details of the respondents were recorded and no sensitive data collected during the interviews. Telephone numbers were at no point connected to name, address or interview content and deleted after the interviews.

**Informed Consent Statement:** Informed consent was obtained from all subjects involved in the study before/during the telephonic interviews.

**Data Availability Statement:** The interview questionnaire, as well as all individual responses and FCMs can be found in the supplementary material for this article.

**Acknowledgments:** The authors are grateful to Sigrun Kabisch and Sven Schneider for their supervision. The authors would also like to extend their gratitude to Sharad Mahajan and Vishnu Shinde from the NGO MASHAL, for connecting the authors to potential interviewees from the four settlements, and two anonymous reviewers for helpful feedback.

**Conflicts of Interest:** The authors declare no conflict of interest.

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
