# Peer review of "Using Fuzzy Cognitive Maps to Assess Liveability in Slum Upgrading Schemes: Case of Pune, India"

_urbansci, doi:10.3390/urbansci5020044_

Round 1

Reviewer 1 Report

Thank you for the opportunity to review this paper, which I thoroughly enjoyed reading. The topic is highly relevant and timely and the authors demonstrate an excellent understanding of their topic, which they present clearly and with excellent figures. Nonetheless, the paper could be improved and I’ve made some suggestions that I hope the authors find helpful.

Section 2, Materials and Methods:

  • Section 2.1’s title describes the interviews as ‘semi-structured’, whereas they are described as ‘structured’ in the first sentence of this sub-section. Which is correct?
  • It is unclear how the semi/structured interviews (as opposed to the questionnaires) were analysed. This should be included in the paper and the interview questions should either be included in the paper or the supplementary information.
  • The authors state that “The first set of questions elicits the interviewees' perception about the overall quality of life, along a three-point scale, from “Satisfactory” to “Can be Improved” to “Unsatisfactory”. However, the questionnaire seems to only have one question related to this (not a set), and that question is “How much do you like your neighbourhood?”. Possible responses are ‘not at all’, ‘a bit’, and ‘a lot’. If this is the ‘set of questions’ to which the authors refer, I argue that the question asked (about liking the neighbourhood) cannot be equated to the responses as reported in the paper (about overall quality of life). These are two quite different things. Additionally, the response options should correlate between the questionnaire and the results reported in the paper.
  • The questionnaires show that participants are asked to rate the influence of indicators as ‘very little’, ‘a bit’ and ‘a lot’, but the paper translates these into ‘very little’, ‘moderate’ and ‘a lot’. This needs making clear to the reader and ideally the paper should be changed to match the questionnaire.
  • The questionnaire includes ‘regular waste collection’ as a liveability indicator, but this does not appear in the paper. The authors need to explain how it correlates with the indicators listed in the paper.
  • The authors omit three indicators from the mapping section of the questionnaire because they are considered to be objectively improved via upgrading (these are: ‘quality of housing’ (should this include ‘structure’?), ‘access to basic amenities’ and ‘security of tenure’). However, I notice that Interviewee 1 of Neighbourhood 1 (Figure 7) rates as ‘satisfactory’ both ‘security of tenure’ and ‘quality of housing structure’. This raises the question as to whether it is reasonable to exclude these three elements from the second phase of the questionnaire. I appreciate that the study cannot be redone with this in mind, so I suggest a discussion of this tension within the paper.
  • Section 2.2.1 lines 315-316 state that degree centrality is the cumulative strength, but clearly it isn’t and it is later accurately described as an average.

Section 3. Results:

  • The results section presents only the findings from the questionnaire. It should also include findings from the semi/structured interviews.
  • The description of Neighbourhood 1 includes the duration of time the interviewees had lived in the settlement and how their housing has changed. I would like to see this level of detail in the descriptions of the other two neighbourhoods, especially as residence duration is mentioned later in the paper.
  • Section 3.2 refers to the ‘okay’ response from the questionnaire for the first time (it also appears on pg 14). Because the response options for this section of the questionnaire don’t correlate with what is reported in the paper (or used to describe the responses for the previous neighbourhood in the paper), this jars and the reader is left wondering what it means.
  • In Section 3.2 the authors make a series of statements about how improving indicator performance can be used to demonstrate “a positive influence on the entire network”, and “directly improves” (lines 392-395). These statements are unsupported by the study. At most, the authors can state that the interviewed residents perceive such relationships.
  • In Section 3.3 line 408 the authors refer to “recurrent complaints”. I found the use of the term ‘complaints’ confusing, as I didn’t see the study set within this context.
  • Section 3.4 makes reference to Figure 15, but I think this should be Figure 14.

Section 4. Discussion:

  • I am very confused as to why the authors have concluded that the social dimensions are the most influential (in the first paragraph of the section), having just presented four key central indicators, three of which are from the physical dimension and only one is from the social dimension. If this conclusion is based upon different data (perhaps that presented in Figure 15), this needs to be made clear and fully described.
  • I was also confused by this sentence, “Although the interviewees from neighbourhood 4 mentioned a decrease in Sense of Belongingness or Good Relationship with Neighbours/Community and the indicators related to community ties were central in 3 out of 4 aggregated FCMs, a definite answer to whether a loss of existing community ties due to in-situ upgradation or relocation would require further investigation” (lines 439-443). In particular, I don’t understand the use of ‘or’ in “Sense of Belongingness or Good Relationship” – should it be ‘and’? Also, what is meant by “the indicators related to community ties”? No indicators specifically use this term and there is no explanation as to what would constitute a grouping of indicators under this description. Finally, it appears to be a sentence fragment.
  • In lines 443-448 the authors discuss that five of their interviewees perceived an improvement in overall liveability. It isn’t clear what data support this claim. Should it be ‘overall quality of life’ (setting aside my previously-mentioned concerns about where the data for ‘overall quality of life’ has come from)? If not, the supporting data should be briefly described.
  • Also in lines 443-448, I note that five interviewees is less than half of all the interviewees and, as such, I’m not convinced the conclusion that this goes against what is said in other literatures is fully defensible.
  • In lines 459-461 the authors seemingly being to draw upon data from the semi/structured interviews. However, this isn’t completely clear because this analysis hasn’t been described in the Methods section nor can the reader see the interview questions.
  • In line 482 the authors make reference to the spatial layout of the neighbourhoods. It would be nice to see this aspect of each neighbourhood described in Section 3. The Google maps are helpful, but cannot show this level of detail or interpretation.
  • Figure 15 is not introduced or explained and, for me, this should be the very first element of Section 4. It would help me greatly in interpreting the results.
  • Figure 15 would benefit from a key explaining to what ‘In-1’ refers (Interviewee 1, of course, but not immediately evident in a paper that contains a lot of acronyms).
  • I’d have liked to have seen the centrality strength measurements for both the individual and aggregated indicators in Figure 15.
  • For the aggregated indicators in Figure 15 (in red), the authors have two instances of two indicators per neighbourhood. In the first instance they use a ‘/’ to join the two, indicating they are combined. In the second instance they use a comma, indicating they are separate. I’d like clarity on this (and the inclusion of centrality strengths would be helpful here).

Supplementary information:

  • In the section of the questionnaire where participants are asked to rate the influence of indicators as ‘very little’, ‘a bit’ and ‘a lot’ the results show four (not three) columns with responses in all four columns. It is unclear what the fourth column is showing.

The figures:

  • The tables in Figures 8, 10, 12 and 14 are a little difficult to read.
